# STOCHASTIC ACTIVATION PRUNING FOR ROBUST ADVERSARIAL DEFENSE

**Guneet S. Dhillon**[1,2]**, Kamyar Azizzadenesheli**[3]**, Zachary C. Lipton**[1,4]**,**
**Jeremy Bernstein**[1,5]**, Jean Kossaifi**[1,6]**, Aran Khanna**[1]**, Anima Anandkumar**[1,5]
[1]Amazon AI, [2]UT Austin, [3]UC Irvine, [4]CMU, [5]Caltech, [6]Imperial College London
`guneetdhillon@utexas.edu, kazizzad@uci.edu, zlipton@cmu.edu,`
`bernstein@caltech.edu, jean.kossaifi@imperial.ac.uk,`
`aran@arankhanna.com, anima@amazon.com`

## ABSTRACT

Neural networks are known to be vulnerable to adversarial examples. Carefully chosen perturbations to real images, while imperceptible to humans, induce misclassification and threaten the reliability of deep learning systems in the wild. To guard against adversarial examples, we take inspiration from game theory and cast the problem as a minimax zero-sum game between the adversary and the model. In general, for such games, the optimal strategy for both players requires a stochastic policy, also known as a *mixed strategy*. In this light, we propose *Stochastic Activation Pruning* (SAP), a mixed strategy for adversarial defense. SAP prunes a random subset of activations (preferentially pruning those with smaller magnitude) and scales up the survivors to compensate. We can apply SAP to pretrained networks, including adversarially trained models, without fine-tuning, providing robustness against adversarial examples. Experiments demonstrate that SAP confers robustness against attacks, increasing accuracy and preserving calibration.

## 1 INTRODUCTION

While deep neural networks have emerged as dominant tools for supervised learning problems, they remain vulnerable to adversarial examples (Szegedy et al., 2013). Small, carefully chosen perturbations to input data can induce misclassification with high probability. In the image domain, even perturbations so small as to be imperceptible to humans can fool powerful convolutional neural networks (Szegedy et al., 2013; Goodfellow et al., 2014). This fragility presents an obstacle to using machine learning in the wild. For example, a vision system vulnerable to adversarial examples might be fundamentally unsuitable for a computer security application. Even if a vision system is not explicitly used for security, these weaknesses might be critical. Moreover, these problems seem unnecessary. If these perturbations are not perceptible to people, why should they fool a machine?

Since this problem was first identified, a rapid succession of papers have proposed various techniques both for *generating* and for *guarding against* adversarial attacks. Goodfellow et al. (2014) introduced a simple method for quickly producing adversarial examples called *the fast gradient sign method* (FGSM). To produce an adversarial example using FGSM, we update the inputs by taking one step in the direction of the *sign* of the gradient of the loss with respect to the input.

To defend against adversarial examples some papers propose training the neural network on adversarial examples themselves, either using the same model (Goodfellow et al., 2014; Madry et al., 2017), or using an ensemble of models (Tramèr et al., 2017a). Taking a different approach, Nayebi & Ganguli (2017) draws inspiration from biological systems. They propose that to harden neural networks against adversarial examples, one should learn flat, compressed representations that are sensitive to a minimal number of input dimensions.

This paper introduces *Stochastic Activation Pruning* (SAP), a method for guarding pretrained networks against adversarial examples. During the forward pass, we stochastically prune a subset of the activations in each layer, preferentially retaining activations with larger magnitudes. Following the pruning, we scale up the surviving activations to normalize the dynamic range of the inputs to the

subsequent layer. Unlike other adversarial defense methods, our method can be applied post-hoc to pretrained networks and requires no additional fine-tuning.

## 2 PRELIMINARIES

We denote an $n$-layered neural network $h : \mathcal{X} \to Y$ as a chain of functions $h = h^n \circ h^{n-1} \circ \ldots \circ h^1$, where each $h^i$ consists of a linear transformation $W^i$ followed by a non-linearity $\phi^i$. Given a set of nonlinearities and weight matrices, a neural network provides a nonlinear mapping from inputs $x \in \mathcal{X}$ to outputs $\hat{y} \in \mathcal{Y}$, i.e.

$$\hat{y} := h(x) = \phi^n(W^n \phi^{n-1}(W^{n-1} \phi^{n-2}(\ldots \phi^1(W^1 x)))).$$

In supervised classification and regression problems, we are given a data set $\mathcal{D}$ of pairs $(x, y)$, where each pair is drawn from an unknown joint distribution. For the classification problems, $y$ is a categorical random variable, and for regression, $y$ is a real-valued vector. Conditioned on a given dataset, network architecture, and a loss function such as cross entropy, a learning algorithm, e.g. stochastic gradient descent, learns parameters $\theta := \{W^i\}_{i=1}^n$ in order to minimize the loss. We denote $J(\theta, x, y)$ as the loss of a learned network, parameterized by $\theta$, on a pair of $(x, y)$. To simplify notation, we focus on classification problems, although our methods are broadly applicable.

Consider an input $x$ that is correctly classified by the model $h$. An adversary seeks to apply a small additive perturbation, $\Delta x$, such that $h(x) \neq h(x + \Delta x)$, subject to the constraint that the perturbation is imperceptible to a human. For perturbations applied to images, the $l_\infty$-norm is considered a better measure of human perceptibility than the more familiar $l_2$ norm Goodfellow et al. (2014). Throughout this paper, we assume that the manipulative power of the adversary, the perturbation $\Delta x$, is of bounded norm $\|\Delta x\|_\infty \leq \lambda$. Given a classifier, one common way to generate an adversarial example is to perturb the input in the direction that increases the cross-entropy loss. This is equivalent to minimizing the probability assigned to the true label. Given the neural network $h$, network parameters $\theta$, input data $x$, and corresponding true output $y$, an adversary could create a perturbation $\Delta x$ as follows

$$\Delta x = \arg \max_{\|r\|_\infty \leq \lambda} J(\theta, x + r, y), \tag{1}$$

Due to nonlinearities in the underlying neural network, and therefore of the objective function $J$, the optimization Eq. 1, in general, can be a non-convex problem. Following Madry et al. (2017); Goodfellow et al. (2014), we use the first order approximation of the loss function

$$\Delta x = \arg \max_{\|r\|_\infty \leq \lambda} [J(\theta, x, y) + r^\top \mathcal{J}(\theta, x, y)], \qquad \text{where } \mathcal{J} = \nabla_x J.$$

The first term in the optimization is not a function of the adversary perturbation, therefore reduces to

$$\Delta x = \arg \max_{\|r\|_\infty \leq \lambda} r^\top \mathcal{J}(\theta, x, y).$$

An adversary chooses $r$ to be in the direction of sign of $\mathcal{J}(\theta, x, y)$, i.e. $\Delta x = \lambda \cdot \text{sign}(\mathcal{J}(\theta, x, y))$. This is the FGSM technique due to Goodfellow et al. (2014). Note that FGSM requires an adversary to access the model in order to compute the gradient.

## 3 STOCHASTIC ACTIVATION PRUNING

Consider the defense problem from a game-theoretic perspective (Osborne & Rubinstein, 1994). The adversary designs a policy in order to maximize the defender's loss, while knowing the defenders policy. At the same time defender aims to come up with a strategy to minimize the maximized loss. Therefore, we can rewrite Eq. 1 as follows

$$\pi^* , \; \rho^* := \arg \min_\pi \max_\rho \mathbb{E}_{p \sim \pi, r \sim \rho} \left[ J(M_p(\theta), x + r, y) \right], \tag{2}$$

where $\rho$ is the adversary's policy, which provides $r \sim \rho$ in the space of bounded (allowed) perturbations (for any $r$ in range of $\rho$, $\|r\|_\infty \leq \lambda$) and $\pi$ is the defenders policy which provides $p \sim \pi$, an instantiation of its policy. The adversary's goal is to maximize the loss of the defender by perturbing

---

**Algorithm 1** Stochastic Activation Pruning (SAP)

---

1: **Input:** input datum $x$, neural network with $n$ layers, with $i^{th}$ layer having weight matrix $W^i$, non-linearity $\phi^i$ and number of samples to be drawn $r^i$.
2: $h^0 \leftarrow x$
3: **for each** layer $i$ **do**
4:      $h^i \leftarrow \phi^i(W^i h^{i-1})$                  ▷ activation vector for layer $i$ with dimension $a^i$
5:      $p_j^i \leftarrow \frac{|(h^i)_j|}{\sum_{k=1}^{a^i} |(h^i)_k|}, \ \forall j \in \{1, \ldots, a^i\}$      ▷ activations normalized on to the simplex
6:      $S \leftarrow \{\}$                                ▷ set of indices not to be pruned
7:      **repeat** $r^i$ **times**              ▷ the activations have $r^i$ chances of being kept
8:          Draw $s \sim \text{categorical}(p^i)$              ▷ draw an index to be kept
9:          $S \leftarrow S \cup \{s\}$                   ▷ add index $s$ to the keep set
10:     **for each** $j \notin S$ **do**
11:        $(h^i)_j \leftarrow 0$                   ▷ prune the activations not in $S$
12:     **for each** $j \in S$ **do**
13:        $(h^i)_j \leftarrow \frac{(h^i)_j}{1-(1-p_j^i)^{r^i}}$         ▷ scale up the activations in $S$
14: **return** $h^n$

---

the input under a strategy $\rho$ and the defender's goal is to minimize the loss by changing model parameters $\theta$ to $M_p(\theta)$ under strategy $\pi$. The optimization problem in Eq. 2 is a *minimax* zero-sum game between the adversary and defender where the optimal strategies $(\pi^*, \rho^*)$, in general, are mixed Nash equilibrium, i.e. stochastic policies.

Intuitively, the idea of SAP is to stochastically drop out nodes in each layer during forward propagation. We retain nodes with probabilities proportional to the magnitude of their activation and scale up the surviving nodes to preserve the dynamic range of the activations in each layer. Empirically, the approach preserves the accuracy of the original model. Notably, the method can be applied post-hoc to already-trained models.

Formally, assume a given pretrained model, with activation layers (*ReLU, Sigmoid, etc.*) and input pair of $(x, y)$. For each of those layers, SAP converts the activation map to a multinomial distribution, choosing each activation with a probability proportional to its absolute value. In other words, we obtain the multinomial distribution of each activation layer with $L_1$ normalization of their absolute values onto a $L_1$-ball simplex. Given the $i$'th layer activation map, $h^i \in \mathbb{R}^{a^i}$, the probability of sampling the $j$'th activation with value $(h^i)_j$ is given by

$$p_j^i = \frac{|(h^i)_j|}{\sum_{k=1}^{a^i} |(h^i)_k|}.$$

We draw random samples with replacement from the activation map given the probability distribution described above. This makes it convenient to determine whether an activation would be sampled at all. If an activation is sampled, we scale it up by the inverse of the probability of sampling it over all the draws. If not, we set the activation to $0$. In this way, SAP preserves inverse propensity scoring of each activation. Under an instance $p$ of policy $\pi$, we draw $r_p^i$ samples with replacement from this multinomial distribution. The new activation map, $M_p(h^i)$ is given by

$$M_p(h^i) = h^i \odot m_p^i, \qquad\qquad (m_p^i)_j = \frac{\mathbb{I}((h^i)_j)}{1-(1-p_j^i)^{r_p^i}},$$

where $\mathbb{I}((h^i)_j)$ is the indicator function that returns 1 if $(h^i)_j$ was sampled at least once, and 0 otherwise. The algorithm is described in Algorithm 1. In this way, the model parameters are changed from $\theta$ to $M_p(\theta)$, for instance $p$ under policy $\pi$, while the reweighting $1 - (1 - p_j^i)^{r_p^i}$ preserves $\mathbb{E}_{p \sim \pi}[M_p(h^i)_j] = (h^i)_j$. If the model was linear, the proposed pruning method would behave the same way as the original model in expectation. In practice, we find that even with the non-linearities in deep neural networks, for sufficiently many examples, SAP performs similarly to the un-pruned model. This guides our decision to apply SAP to pretrained models without performing fine-tuning.

### 3.1 ADVANTAGE AGAINST ADVERSARIAL ATTACK

We attempt to explain the advantages of SAP under the assumption that we are applying it to a pre-trained model that achieves high generalization accuracy. For instance $p$ under policy $\pi$, if the number of samples drawn for each layer $i$, $r_p^i$, is large, then fewer parameters of the neural network are pruned, and the scaling factor gets closer to $1$. Under this scenario, the stochastically pruned model performs almost identically to the original model. The stochasticity is not advantageous in this case, but there is no loss in accuracy in the pruned model as compared to the original model.

On the other hand, with fewer samples in each layer, $r_p^i$, a large number of parameters of the neural network are pruned. Under this scenario, the SAP model's accuracy will drop compared to the original model's accuracy. But this model is stochastic and has more freedom to deceive the adversary. So the advantage of SAP comes if we can balance the number of samples drawn in a way that negligibly impacts accuracy but still confers robustness against adversarial attacks.

SAP is similar to the *dropout* technique due to Srivastava et al. (2014). However, there is a crucial difference: SAP is more likely to sample activations that are high in absolute value, whereas dropout samples each activation with the same probability. Because of this difference, SAP, unlike dropout, can be applied post-hoc *to pretrained models* without significantly decreasing the accuracy of the model. Experiments comparing SAP and dropout are included in section 4. Interestingly, dropout confers little advantage over the baseline. We suspect that the reason for this is that the dropout training procedure encourages all possible dropout masks to result in similar mappings.

### 3.2 ADVERSARIAL ATTACK ON SAP

If the adversary knows that our defense policy is to apply SAP, it might try to calculate the best strategy against the SAP model. Given the neural network $h$, input data $x$, corresponding true output $y$, a policy $\rho$ over the allowed perturbations, and a policy $\pi$ over the model parameters that come from SAP (this result holds true for any stochastic policy chosen over the model parameters), the adversary determines the optimal policy $\rho^*$

$$\rho^* = \arg\max_\rho \mathbb{E}_{p\sim\pi, r\sim\rho}[J(M_p(\theta), x + r, y)].$$

Therefore, using the result from section 2, the adversary determines the perturbation $\Delta x$ as follows;

$$\Delta x = \arg\max_r r^\top \mathbb{E}_{p\sim\pi}[\mathcal{J}(M_p(\theta), x, y)].$$

To maximize the term, the adversary will set $r$ to be in the direction of sign of $\mathbb{E}_{p\sim\pi}[\mathcal{J}(M_p(\theta), x, y)]$. Analytically computing $\mathbb{E}_{p\sim\pi}[\mathcal{J}(M_p(\theta), x, y)]$ is not feasible. However, the adversary can use Monte Carlo (MC) sampling to estimate the expectation as $\widetilde{\mathcal{J}}(M_p(\theta), x, y)$. Then, using FGSM, $\Delta x = \lambda \cdot \text{sign}(\widetilde{\mathcal{J}}(M_p(\theta), x, y))$.

## 4 EXPERIMENTS

Our experiments to evaluate SAP address two tasks: image classification and reinforcement learning. We apply the method to the *ReLU* activation maps at each layer of the pretrained neural networks. To create adversarial examples in our evaluation, we use FGSM, $\Delta x = \lambda \cdot \text{sign}(\mathcal{J}(M_p(\theta), x, y))$. For stochastic models, the adversary estimates $\mathcal{J}(M_p(\theta), x, y)$ using MC sampling unless otherwise mentioned. All perturbations are applied to the pixel values of images, which normally take values in the range 0-255. So the fraction of perturbation with respect to the data's dynamic range would be $\frac{\lambda}{256}$. To ensure that all images are valid, even following perturbation, we clip the resulting pixel values so that they remain within the range $[0, 255]$. In all plots, we consider perturbations of the following magnitudes $\lambda = \{0, 1, 2, 4, 8, 16, 32, 64\}$.[1]

To evaluate models in the image classification domain, we look at two aspects: the model accuracy for varying values of $\lambda$, and the calibration of the models (Guo et al., 2017). Calibration of a model is the relation between the confidence level of the model's output and its accuracy. A linear calibration

---

[1]All the implementations were coded in MXNet framework (Chen et al., 2015) and sample code is available at `https://github.com/Guneet-Dhillon/Stochastic-Activation-Pruning`

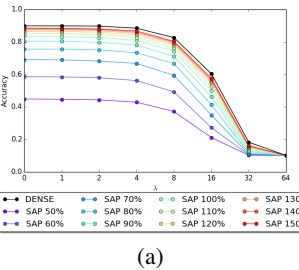 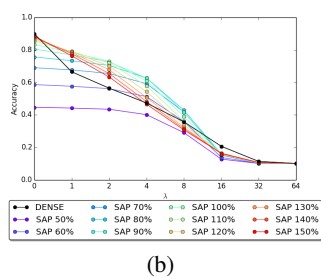 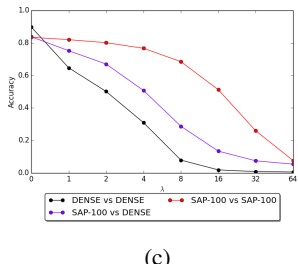

(a)  (b)  (c)

Figure 1: Accuracy plots of a variety of attacks against dense model and SAP models with different perturbation strengths, $\lambda$. For the SAP $\tau\%$ models, $\tau$ denotes the percentages of samples drawn from the multinomial distribution, at each layer. (a) SAP models tested against random perturbation. (b) SAP models tested against the FGSM attack, using MC sampling. (c) SAP-100 tested against an iterative adversarial attack, using MC sampling (legend shows defender vs. adversary). It is worth restating that obtaining the iterative attack of SAP models is much more expensive and noisier than the iterative attack of dense models.

is ideal, as it suggests that the accuracy of the model is proportional to the confidence level of its output. To evaluate models in the reinforcement learning domain, we look at the average score that each model achieves on the games played, for varying values of $\lambda$. The higher the score, the better is the model's performance. Because the units of reward are arbitrary, we report results in terms of the the relative percent change in rewards. In both cases, the output of stochastic models are computed as an average over multiple forward passes.

## 4.1 ADVERSARIAL ATTACKS IN IMAGE CLASSIFICATION

The CIFAR-10 dataset (Krizhevsky & Hinton, 2009) was used for the image classification domain. We trained a ResNet-20 model (He et al., 2016) using SGD, with minibatches of size 512, momentum of 0.9, weight decay of 0.0001, and a learning rate of 0.5 for the first 100 epochs, then 0.05 for the next 30 epochs, and then 0.005 for the next 20 epochs. This achieved an accuracy of 89.8% with cross-entropy loss and *ReLU* non-linearity. For all the figures in this section, we refer to this model as the *dense* model. The accuracy of the dense model degrades quickly with $\lambda$. For $\lambda = 1$, the accuracy drops down to 66.3%, and for $\lambda = 2$ it is 56.4%. These are small (hardly perceptible) perturbations in the input images, but the dense model's accuracy decreases significantly.

### 4.1.1 STOCHASTIC ACTIVATION PRUNING (SAP)

We apply SAP to the dense model. For each activation map $h^i \in \mathbb{R}^{a^i}$, we pick $k\%$ of $a^i$ activations to keep. Since activations are sampled with replacement, $k$ can be more than 100. We will refer to $k$ as the percentage of samples drawn. Fig. 1a plots performance of SAP models against examples perturbed with random noise. Perturbations of size $\lambda = 64$ are readily perceptible and push all models under consideration to near-random outputs, so we focus our attention on smaller values of $\lambda$. Fig. 1b plots performance of these models against adversarial examples. With many samples drawn, SAP converges to the dense model. With few samples drawn, accuracy diminishes for $\lambda = 0$, but is higher for $\lambda \neq 0$. The plot explains this balance well. We achieve the best performance with $\sim 100\%$ samples picked. We will now only look at SAP $100\%$ (SAP-100). Against adversarial examples, with $\lambda = 1, 2$ and $4$, we observe a 12.2%, 16.3% and 12.8% absolute increase in accuracy respectively. However, for $\lambda = 0$, we observe a 6.5% absolute decrease in accuracy. For $\lambda = 16$ again, there is a 5.2% absolute decrease in accuracy.

### 4.1.2 DROPOUT (DRO)

*Dropout*, a technique due to Srivastava et al. (2014), was also tested to compare with SAP. Similar to the SAP setting, this method was added to the *ReLU* activation maps of the dense model. We see that low dropout rate perform similar to the dense model for small $\lambda$ values, but its accuracy starts decreasing very quickly for higher $\lambda$ values (Fig. 2a). We also trained ResNet-20 models, similar

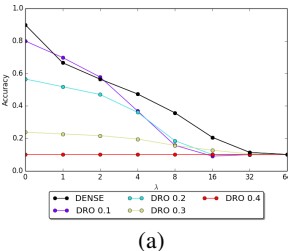 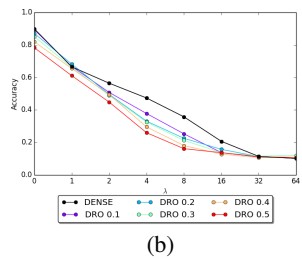 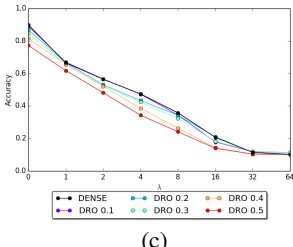

|  (a)  |  (b)  |  (c)  |

Figure 2: Robustness of dropout models, with different rates of dropout (denoted in the legends), against adversarial attacks, using MC sampling, with a variety of perturbation strengths, $\lambda$: $(a)$ dropout is applied on the pre-trained models during the validations; $(b)$ the models are trained using dropout, and dropout is applied during the validations; $(c)$ the models are trained using dropout, but dropout is not applied during the validations.

to the dense model, but with different dropout rates. This time, the models were trained for 250 epochs, with an initial learning rate of 0.5, reduced by a factor of 0.1 after 100, 150, 190 and 220 epochs. These models were tested against adversarial examples with and without dropout during validation (Figs. 2b and 2c respectively). The models do similar to the dense model, but do not provide additional robustness.

### 4.1.3 ADVERSARIAL TRAINING (ADV)

*Adversarial training* (Goodfellow et al., 2014) has emerged a standard method for defending against adversarial examples. It has been adopted by Madry et al. (2017); Tramèr et al. (2017a) to maintain high accuracy levels even for large $\lambda$ values. We trained a ResNet-20 model, similar to the dense model, with an initial learning rate of 0.5, which was halved every 10 epochs, for a total of 100 epochs. It was trained on a dataset consisting of $80\%$ un-perturbed data and $20\%$ adversarially perturbed data, generated on the model from the previous epoch, with $\lambda = 2$. This achieved an accuracy of $75.0\%$ on the un-perturbed validation set. Note that the model capacity was not changed. When tested against adversarial examples, the accuracy dropped to $72.9\%, 70.9\%$ and $67.5\%$ for $\lambda = 1, 2$ and $4$ respectively. We ran SAP-100 on the ADV model (referred to as ADV+SAP-100). The accuracy in the no perturbation case was $74.1\%$. For adversarial examples, both models act similar to each other for small values of $\lambda$. But for $\lambda = 16$ and $32$, ADV+SAP-100 gets a higher accuracy than ADV by an absolute increase of $7.8\%$ and $7.9\%$ respectively.

We compare the accuracy-$\lambda$ plot for dense, SAP-100, ADV and ADV+SAP-100 models. This is illustrated in Fig. 3 For smaller values of $\lambda$, SAP-100 achieves high accuracy. As $\lambda$ gets larger, ADV+SAP-100 performs better than all the other models. We also compare the calibration plots for these models, in Fig. 4. The dense model is not linear for any $\lambda \neq 0$. The other models are well calibrated (close to linear), and behave similar to each other for $\lambda \leq 4$. For higher values of $\lambda$, we see that ADV+SAP-100 is the closest to a linearly calibrated model.

### 4.2 ADVERSARIAL ATTACKS IN DEEP REINFORCEMENT LEARNING (RL)

Previously, (Behzadan & Munir, 2017; Huang et al., 2017; Kos & Song, 2017) have shown that the reinforcement learning agents can also be easily manipulated by adversarial examples. The RL agent learns the long term value $Q(a, s)$ of each state-action pair $(s, a)$ through interaction with an environment, where given a state $s$, the optimal action is $\arg\max_a Q(a, s)$. A regression based algorithm, Deep Q-Network (DQN)(Mnih et al., 2015) and an improved variant, Double DQN (DDQN) have been proposed for the popular Atari games (Bellemare et al., 2013) as benchmarks. We deploy DDQN algorithm and train an RL agent in variety of different Atari game settings.

Similar to the image classification experiments, we tested SAP on a pretrained model (the model is described in the Appendix section A), by applying the method on the *ReLU* activation maps. SAP-100 was used for these experiments. Table 1 specifies the relative percentage increase in rewards of SAP-100 as compared to the original model. For all the games, we observe a drop in performance for

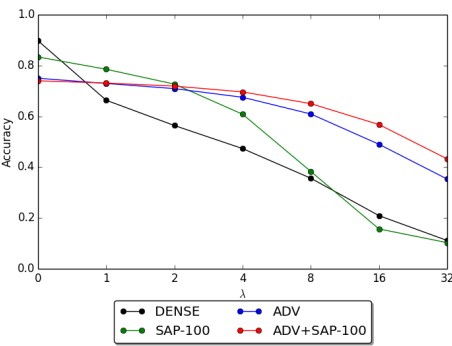

Figure 3: Accuracy plots of the dense, SAP-100, ADV and ADV+SAP-100 models, against adversarial attacks, using MC sampling, with a variety of perturbation strengths, $\lambda$.

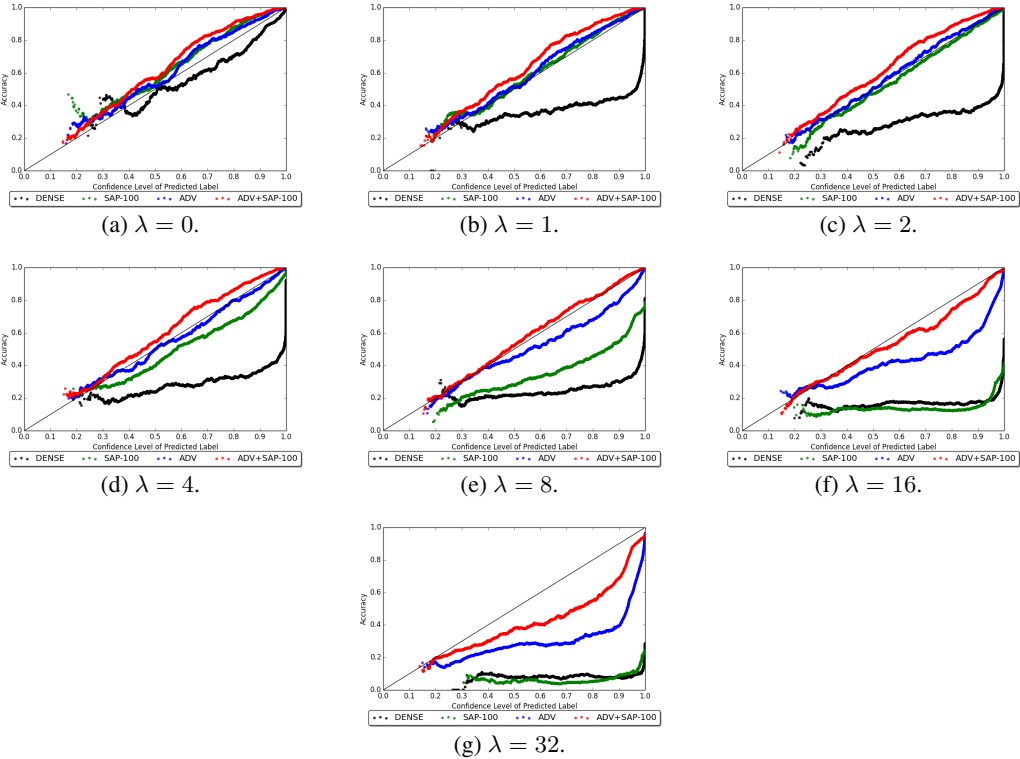

Figure 4: Calibration plots of the dense, SAP-100, ADV and ADV+SAP-100 models, against adversarial attacks, using MC sampling, with a variety of different perturbation strengths, $\lambda$. These plots show the relation between the confidence level of the model's output and its accuracy.

the no perturbation case. But for $\lambda \neq 0$, the relative increase in rewards is positive (except for $\lambda = 1$ in the BattleZone game), and is very high in some cases (3425.9% for $\lambda = 1$ for the Bowling game).

## 4.3 ADDITIONAL BASELINES

In addition to experimenting with SAP, dropout, and adversarial training, we conducted extensive experiments with other methods for introducing stochasticity into a neural network. These techniques included 0-mean Gaussian noise added to weights (RNW), 1-mean multiplicative Gaussian noise for the weights (RSW), and corresponding additive (RNA) and multiplicative (RSA) noise added to

Table 1: Relative percentage increase in rewards gained for SAP-100 compared to original model while playing different Atari games.

| $\lambda$ | Assault | Asterix | BankHeist | BattleZone | BeamRider | Bowling |
|---|---|---|---|---|---|---|
| 0 | -12.2% | -33.4% | -59.2% | -65.8% | -15.8% | -4.5% |
| 1 | 10.4% | 13.3% | 131.7% | -22.0% | 164.5% | 3425.9% |
| 2 | 9.8% | 20.8% | 204.8% | 110.1% | 92.3% | |
| 4 | 12.4% | 14.0% | 1760.0% | 202.6% | | |
| 8 | 16.6% | 7.4% | 60.9% | 134.8% | | |

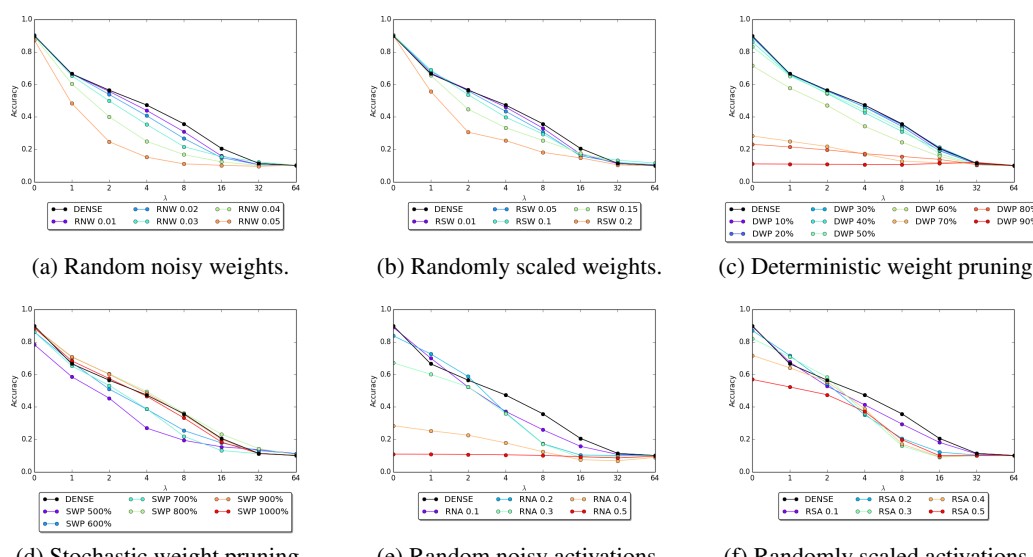

(a) Random noisy weights.

(b) Randomly scaled weights.

(c) Deterministic weight pruning.

(d) Stochastic weight pruning.

(e) Random noisy activations.

(f) Randomly scaled activations.

Figure 5: Robustness of different pruning and noisifying strategies against their respective adversarial attacks (MC sampling used to estimate gradients of stochastic models).

the activations. We describe each method in detail in Appendix B. Each of these models performs worse than the dense baseline at most levels of perturbation and none matches the performance of SAP. Precisely why SAP works while other methods introducing stochasticity do not, remains an open question that we continue to explore in future work.

### 4.4 SAP ATTACKS WITH VARYING NUMBERS OF MC SAMPLES

In the previous experiments the SAP adversary used 100 MC samples to estimate the gradient. Additionally, we compared the performance of SAP-100 against various attacks, these include the standard attack calculated based on the dense model and those generated on SAP-100 by estimating the gradient with various numbers of MC samples. We see that if the adversary uses the dense model to generate adversarial examples, SAP-100 model's accuracy decreases. Additionally, if the adversary uses the SAP-100 model to generate adversarial examples, greater numbers of MC samples lower the accuracy more. Still, even with 1000 MC samples, for low amounts of perturbation ($\lambda = 1$ and 2), SAP-100 retains higher accuracy than the dense model.

Computing a single backward pass of the SAP-100 model for 512 examples takes $\sim 20$ seconds on 8 GPUs. Using 100 and 1000 MC samples would take $\sim 0.6$ and $\sim 6$ hours respectively.

### 4.5 ITERATIVE ADVERSARIAL ATTACK

A more sophisticated technique for producing adversarial perturbations (than FGSM) is to apply multiple and smaller updates to the input in the direction of the local sign-gradients. This can be done by taking small steps of size $k \leq \lambda$ in the direction of the sign-gradient at the updated point and

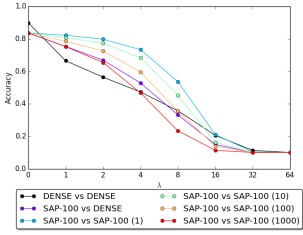

Figure 6: Accuracy plots of adversarial attacks, with different perturbation strengths, $\lambda$. The legend shows defender vs. adversary models (used for gradient computation), and the number of MC samples used to estimate the gradient.

repeating the procedure $\lceil \frac{\lambda}{k} \rceil$ times (Kurakin et al., 2016) as follows

$$x^0 = x \ , \qquad x^t = clip_{x,\lambda}\left(x^{t-1} + k\,\mathrm{sign}(\mathcal{J}(\theta, x^{t-1}, y))\right),$$

where function $clip_{x,\lambda}$ is a projection into a $L_\infty$-ball of radius $\lambda$ centered at $x$, and also into the hyper-cube of image space (each pixel is clipped to the range of $[0, 255]$). The dense and SAP-100 models are tested against this adversarial attack, with $k = 1.0$ (Fig. 1c). The accuracies of the dense model at $\lambda = 0, 1, 2$ and 4 are 89.8%, 66.3%, 50.1% and 31.0% respectively. The accuracies of the SAP-100 model against attacks computed on the same model (with 10 MC samples taken at each step to estimate the gradient) are 83.3%, 82.0%, 80.2% and 76.7%, for $\lambda = 0, 1, 2, 4$ respectively. The SAP-100 model provides accuracies of 83.3%, 75.2%, 67.0% and 50.8%, against attacks computed on the dense model, with the perturbations $\lambda = 0, 1, 2, 4$ respectively. Iterative attacks on the SAP models are much more expensive to compute and noisier than iterative attacks on dense models. This is why the adversarial attack computed on the dense model results in lower accuracies on the SAP-100 model than the adversarial attack computed on the SAP-100 model itself.

## 5 RELATED WORK

Robustness to adversarial attack has recently emerged as a serious topic in machine learning (Goodfellow et al., 2014; Kurakin et al., 2016; Papernot & McDaniel, 2016; Tramèr et al., 2017b; Fawzi et al., 2018). Goodfellow et al. (2014) introduced FGSM. Kurakin et al. (2016) proposed an iterative method where FGSM is used for smaller step sizes, which leads to a better approximation of the gradient. Papernot et al. (2017) observed that adversarial examples could be transferred to other models as well. Madry et al. (2017) introduce adding random noise to the image and then using the FGSM method to come up with adversarial examples.

Being robust against adversarial examples has primarily focused on training on the adversarial examples. Goodfellow et al. (2014) use FGSM to inject adversarial examples into their training dataset. Madry et al. (2017) use an iterative FGSM approach to create adversarial examples to train on. Tramèr et al. (2017a) introduced an ensemble adversarial training method of training on the adversarial examples created on the model itself and an ensemble of other pre-trained models. These works have been successful, achieving only a small drop in accuracy form the clean and adversarially generated data. Nayebi & Ganguli (2017) proposes a method to produce a smooth input-output mapping by using saturating activation functions and causing the activations to become saturated.

## 6 CONCLUSION

The SAP approach guards networks against adversarial examples without requiring any additional training. We showed that in the adversarial setting, applying SAP to image classifiers improves both the accuracy and calibration. Notably, combining SAP with adversarial training yields additive benefits. Additional experiments show that SAP can also be effective against adversarial examples in reinforcement learning.

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

## A  REINFORCEMENT LEARNING MODEL ARCHITECTURE

For the experiments in section 4.2, we trained the network with RMSProp, minibatches of size 32, a learning rate of 0.00025, and a momentum of 0.95 and as in (Mnih et al., 2015) where the discount factor is $\gamma = 0.99$, the number of steps between target updates to 10000 steps. We updated the network every 4 steps by randomly sampling a minibatch of size 32 samples from the replay buffer and trained the agents for a total of $100M$ steps per game. The experience replay contains the $1M$ most recent transitions. For training we used an $\varepsilon$-greedy policy with $\varepsilon$ annealed linearly from 1 to $0.1$ over the first $1M$ time steps and fixed at $0.1$ thereafter.

The input to the network is $4 \times 84 \times 84$ tensor with a rescaled, gray-scale version of the last four observations. The first convolution layer has 32 filters of size 8 with a stride of 4. The second convolution layer has 64 filters of size 4 with stride 2. The last convolution layer has 64 filters of size 3 followed by two fully connected layers with size 512 and the final fully connected layer Q-value of each action where *ReLU* rectifier is deployed for the nonlinearity at each layer.

## B  OTHER METHODS

We tried a variety of different methods that could be added to pretrained models and tested their performance against adversarial examples. The following is a continuation of section 4.1, where we use the dense model again on the CIFAR-10 dataset.

### B.1  RANDOM NOISY WEIGHTS (RNW)

One simple way of introducing stochasticity to the activations is by adding random Gaussian noise to each weight, with mean 0 and constant standard deviation, $s$. So each weight tensor $W^i$ now changes to $M(W^i)$, where the $j$'th entry is given by

$$M(W^i)_j = (W^i)_j + \eta, \qquad\qquad \eta \sim \mathcal{N}(0, s^2).$$

These models behave very similar to the dense model (Fig. 5a, the legend indicates the value of $s$). While we test several different values of $s$, we do not observe any significant improvements regarding robustness against adversarial examples. As $s$ increased, the accuracy for non-zero $\lambda$ decreased.

### B.2  RANDOMLY SCALED WEIGHTS (RSW)

Instead of using additive noise, we also try multiplicative noise. The scale factor can be picked from a Gaussian distribution, with mean 1 and constant standard deviation $s$. So each weight tensor $W^i$ now changes to $M(W^i)$, where the $j$'th entry is given by

$$M(W^i)_j = \eta \cdot (W^i)_j, \qquad\qquad \eta \sim \mathcal{N}(1, s^2).$$

These models perform similar to the dense model, but again, no robustness is offered against adversarial examples. They follow a similar trend as the RNW models (Figure 5b, the legend indicates the value of $s$).

### B.3  DETERMINISTIC WEIGHT PRUNING (DWP)

Following from the motivation of preventing perturbations to propagate forward in the network, we tested deterministic weight pruning, where the top $k\%$ entries of a weight matrix were kept, while the rest were pruned to 0, according to their absolute values. This method was prompted by the success achieved by this pruning method, introduced by Han et al. (2015), where they also fine-tuned the model.

For low levels of pruning, these models do very similar to the dense model, even against adversarial examples (Fig. 5c, the legend indicates the value of $k$). The adversary can compute the gradient of the sparse model, and the perturbations propagate forward through the surviving weights. For higher levels of sparsity, the accuracy in the no-perturbation case drops down quickly.

## B.4 STOCHASTIC WEIGHT PRUNING (SWP)

Observing the failure of deterministic weight pruning, we tested a mix of stochasticity and pruning, the stochastic weight pruning method. Very similar to the idea of SAP, we consider all the entries of a weight tensor to be a multinomial distribution, and we sample from it with replacement. For a weight tensor $W^i \in \mathbb{R}^{a^i}$, we sample from it $r^i$ times with replacement. The probability of sampling $(W^i)_j$ is given by

$$p_j^i = \frac{|(W^i)_j|}{\sum_{k=1}^{a^i} |(W^i))_k|}.$$

The new weight entry, $M(W^i)_j$, is given by

$$M(W^i)_j = W_j^i \cdot \frac{\mathbb{I}(W_j^i)}{1 - (1 - p_i)^{r^i}},$$

where $\mathbb{I}(W_j^i)$ is the indicator function that returns 1 if $W_j^i$ was sampled at least once, and 0 otherwise.

For these experiments, for each weight matrix $W^i \in \mathbb{R}^{m^i}$, the number of samples picked were $k\%$ of $m^i$. Since samples were picked with replacement, $k$ could be more than 100. We will refer to $k$ as the percentage of samples drawn.

These models behave very similar to the dense model. We tried drawing range of percentages of samples, but no evident robustness could be seen against adversarial examples (Figure 5d, the legend indicates the value of $k$). For a small $s$, it is very similar to the dense model. As $s$ increases, the these models do marginally better for low non-zero $\lambda$ values, and then drops again (similar to the SAP case).

## B.5 RANDOM NOISY ACTIVATIONS (RNA)

Next we change our attention to the activation maps in the dense model. One simple way of introducing stochasticity to the activations is by adding random Gaussian noise to each activation entry, with mean 0 and constant standard deviation, $s$. So each activation map $h^i$ now changes to $M(h^i)$, where the $j$'th entry is given by

$$M(h^i)_j = (h^i)_j + \eta, \qquad\qquad \eta \sim \mathcal{N}(0, s^2).$$

These models too do not offer any robustness against adversarial examples. Their accuracy drops quickly with $\lambda$ and $s$ (Fig. 5e, the legend indicates the value of $s$).

## B.6 RANDOMLY SCALED ACTIVATIONS (RSA)

Instead of having additive noise, we can also make the model stochastic by scaling the activations. The scale factor can be picked from a Gaussian distribution, with mean 1 and constant standard deviation $s$. So each activation map $h^i$ now changes to $M(h^i)$, where the $j$'th entry is given by

$$M(h^i)_j = \eta(h^i)_j, \qquad\qquad \eta \sim \mathcal{N}(1, s^2).$$

These models perform similar to the dense model, exhibiting no additional robustness against adversarial examples (Figure 5f, the legend indicates the value of $s$).

