# OpenReview forum: "Stochastic Activation Pruning for Robust Adversarial Defense"
_ICLR.cc/2018/Conference — Accept (Poster)_

### Official Review · AnonReviewer2 · 2017-11-13
**overall, this paper presents a practical method to prevent a classifier from adversarial examples, which can be applied in addition to adversarial training. The presentation could be improved.**

**Rating:** 6
**Confidence:** 3

**Review:**

This paper investigates a new approach to prevent a given classifier from adversarial examples. The most important contribution is that the proposed algorithm can be applied post-hoc to already trained networks. Hence, the proposed algorithm (Stochastic Activation Pruning) can be combined with algorithms which prevent from adversarial examples during the training.

The proposed algorithm is clearly described. However there are issues in the presentation.

In section 2-3, the problem setting is not suitably introduced.
In particular one sentence that can be misleading:
“Given a classifier, one common way to generate an adversarial example is to perturb the input in direction of the gradient…”
You should explain that given a classifier with stochastic output, the optimal way to generate an adversarial example is to perturb the input proportionally to the gradient. The practical way in which the adversarial examples are generated is not known to the player. An adversary could choose any policy. The only thing the player knows is the best adversarial policy.

In section 4, I do not understand why the adversary uses only the sign and not also the value of the estimated gradient. Does it come from a high variance? If it is the case, you should explain that the optimal policy of the adversary is approximated by “fast gradient sign method”.

In comparison to dropout algorithm, SAP shows improvements of accuracy against adversarial examples. SAP does not perform as well as adversarial training, but SAP could be used with a trained network.

Overall, this paper presents a practical method to prevent a classifier from adversarial examples, which can be applied in addition to adversarial training. The presentation could be improved.

---

> ### Author Response · Authors · 2017-12-28
> **Response AnonReviewer2**
>
> We thank Reviewer2 for thorough comments. We are glad that the reviewer appreciated the value of an adversarial defense technique that can be applied post-hoc and our exposition. We reply to specific points below:
>
> 1. You are correct, the defender does not know the what policy any actual adversary will use, only what the optimal adversary might do, thus the objective of minimizing the worst-case performance. We are improving the draft to be clearer in this regard.
>
> 2. Regarding: “In section 4, I do not understand why the adversary uses only the sign and not also the value of the estimated gradient.”: The reason why we are considering only the sign is because we cap the infinity norm of the adversarial perturbation. This leads to taking a step of equal size in each input dimension and thus the gradient magnitude does not come into play. This approach is standard in the recent academic study of adversarial examples and follows work by Goodfellow et al. (2014), which showed that imperceptible adversarial examples could be produced efficiently in this manner..
>
> One motivation for considering the infinity norm (vs L2 or L1) for constraining the size of an adversarial perturbation is that it accords more closely with perceptual similarity. For example, it’s possible to devise a perturbation with small L2 norm that is perceptually obvious because it moves a small group of pixels a large amount.
>
> Naturally, a stronger adversary might pursue an iterative approach rather than making one large perturbation. To this end, we are currently running experiments with iterative attacks and the initial results are promising - SAP continues to significantly outperform the dense model. We will add these results to the paper when they are ready.
>
> 3. We are grateful for the reviewer’s suggestions for improving the exposition and are currently working to revise the draft in accordance with these recommendations. To start, we have improved some of the (previously) confusing language that might have failed to distinguish between the optimal adversary and some arbitrary adversary which may not apply the optimal perturbation.

---

### Official Review · AnonReviewer1 · 2017-11-27
**Simple and yet effective method against adversarial attack post traning.**

**Rating:** 7
**Confidence:** 4

**Review:**

This paper propose a simple method for guarding trained models against adversarial attacks. The method is to prune the network’s activations at each layer and renormalize the outputs. It’s a simple method that can be applied post-training and seems to be effective.

The paper is well written and easily to follow. Method description is clear. The analyses are interesting and done well. I am not familiar with the recent work in this area so can not judge if they compare against SOTA methods but they do compare against various other methods.

Could you elaborate more on the findings from Fig 1.c Seems that  the DENSE model perform best against randomly perturbed images. Would be good to know if the authors have any intuition why is that the case.

There are some interesting analysis in the appendix against some other methods, it would be good to briefly refer to them in the main text.

I would be interested to know more about the intuition behind the proposed method. It will make the paper stronger if there were more content arguing analyzing the intuition and insight that lead to the proposed method.

Also would like to see some notes about computation complexity of sampling multiple times from a larger multinomial.

Again I am not familiar about different kind of existing adversarial attacks, the paper seem to be mainly focus on those from Goodfellow et al 2014. Would be good to see the performance against other forms of adversarial attacks as well if they exist.

---

> ### Author Response · Authors · 2017-12-28
> **Response to AnonReviewer1**
>
> Thanks for your clear review of our paper. We are glad that you appreciated both the method and the clarity of exposition.
>
> 1. Regarding Fig 1.c: While dense models are susceptible to adversarial attack, they are actually quite robust to random noise. The purpose of reporting the results of this experiment is to provide context for the other results. Because dense models are not especially vulnerable to random noise, we are not surprised that they perform well here.
>
> 2. Thanks for the suggestion that the analysis in the appendix should be summarized within the body of the paper. Per your request, we have added an additional subsection (5.3) in the current draft that briefly describes the baselines and we have included a corresponding figure that shows the quantitative results for each.
>
> 3. While we are reluctant to present an explanation for a phenomena that we do not fully understand, we are happy to share the intuitions that guided us in developing the algorithm:
>
> Originally we were looking sparsifying the weights and/or activations of the network. We were encouraged by results, e.g. https://arxiv.org/abs/1510.00149, showing high accuracy with sparsified weights (as by pruning). We thought that by sparsifying a network, we might maintain high accuracy while lowering the Lipschitz constant and thus conferring some robustness against small perturbations. We later drew some inspiration from randomized algorithms that sparsify matrices by randomly dropping entries according to their weights and scaling up the survivors to produce a sparse matrix with similar spectral properties to the original.
>
> 4. Sampling from the multinomial is fast. Without getting into detail about how many random bits are needed, given uniform samples, we can convert to a sample from a multinomial by performing a binary search. So it’s roughly k log(n) where k is the number of samples and n is the number of activations. As a practical concern, sampling from the multinomial in our algorithms does not comprise a significant computational obstacle.
>
> 5. As you correctly point out, In our experiments, we adopt approach from Goodfellow et al. of evaluating with adversarial perturbations produced by taking a single step with capped infinity norm. However, we generate these attacks differently for each model. Against our stochastic models, the adversary produces the attack by estimating the gradient with MC samples.
>
> 6. Per your suggestions we have compared against a stronger modes of attack, namely an iterative update where we take multiple small updates, each of capped infinity norm. In these experiments, SAP continues to outperform the dense model significantly. We are currently compiling these results and will add them to the draft when ready.

---

### Official Review · AnonReviewer3 · 2017-11-27
**Interesting heuristic but little theoretical justification**

**Rating:** 6
**Confidence:** 4

**Review:**

The authors propose to improve the robustness of trained neural networks against adversarial examples by randomly zeroing out weights/activations. Empirically the authors demonstrate, on two different task domains, that one can trade off some accuracy for a little robustness -- qualitatively speaking.

On one hand, the approach is simple to implement and has minimal impact computationally on pre-trained networks. On the other hand, I find it lacking in terms of theoretical support, other than the fact that the added stochasticity induces a certain amount of robustness. For example, how does this compare to random perturbation (say, zero-mean) of the weights? This adds stochasticity as well so why and why not this work? The authors do not give any insight in this regard.

Overall, I still recommend acceptance (weakly) since the empirical results may be valuable to a general practitioner. The paper could be strengthened by addressing the issues above as well as including more empirical results (if nothing else).

---

> ### Author Response · Authors · 2017-12-28
> **Response to AnonReviewer3**
>
> Thanks for the thoughtful review of our paper. We are glad that you recognize the empirical strength of the result and the simplicity of the method. We are also share your desire for greater theoretical understanding.
>
> Regarding: “how does this compare to random perturbation (say, zero-mean) of the weights?”.
> We ran this experiment, and found that it did not help. Additionally, for a more direct comparison, we compared against zero-mean Gaussian noise applied to the activations. We call this method Random Noisy Activations (RNA). It was previously described only in Appendix B, but we have now added a brief description to section 5 and reported the quantitative results in Figure 5.
>
> Despite extensive empirical study, precisely why our method works but random noise on the activations does not remains unclear. While we can imagine some ways of spinning a theoretical story post-hoc, the honest answer is that we do not yet possess a solid theoretical explanation. We share your desire for a greater understanding and plan to investigate this direction further in future work.
>
> ***TL;DR: Per your suggestions, we have improved the draft by running additional experiments. Please find in Figure 5 results for 0-mean gaussian noise applied to weights with sigma values {.01, .02, …, .05}, as well as results for several other sensible baselines and greater detail in Appendix B.***

---

### Author Response · Authors · 2017-12-28
**General reply to all reviewers**

We would like to thank the reviewers for their thoughtful responses to our paper. We are glad to see that there is a consensus among the reviewers to accept and are grateful to each of the reviewers for critical suggestions that will help us to improve the work. Please find individual replies to each of the reviews in the respective threads.

---

### Decision · Program_Chairs · 2018-01-29
**ICLR 2018 Conference Acceptance Decision**

**Decision:**

Accept (Poster)

**Comment:**

This is a borderline paper.  The reviewers are happy with the simplicity of the proposed method and the fact that it can be applied after training; but are concerned by the lack of theory explaining the results.  I will recommend accepting, but I would ask the authors add the additional experiments they have promised, and would also suggest experiments on imagenet.